# Rice *TCD8* Encoding a Multi-Domain GTPase Is Crucial for Chloroplast Development of Early Leaf Stage at Low Temperatures

**DOI:** 10.3390/biology11121738

**Published:** 2022-11-29

**Authors:** Dongzhi Lin, Ting Zhang, Yu Chen, Mengjie Fan, Rongrong Kong, Lu Chen, Yulu Wang, Jianlong Xu, Yanjun Dong

**Affiliations:** 1College of Life Sciences, Shanghai Normal University, Shanghai 200234, China; 2The Institute of Crop Sciences, Chinese Academy of Agricultural Sciences, 12 South Zhong-Guan Cun Street, Beijing 100081, China; 3Shanghai Key Laboratory of Plant Molecular Sciences, Shanghai 200234, China

**Keywords:** rice, albino phenotype, chloroplast, low temperature, multi-domain GTPase

## Abstract

**Simple Summary:**

Low temperatures do not favor plant chloroplast development. Although numerous genes have been identified to function in modulating chloroplast development, the function of the multi-domain GTPase under cold stress is still scarce in rice. This study demonstrated the *TCD8* encoding the multi-domain GTPase which plays a part in the development of rice chloroplast under lower temperature, suggesting the significance of the cold-inducible *TCD8* for cold-responsive gene regulation.

**Abstract:**

The multi-domain GTPase (MnmE) is conservative from bacteria to human and participates in tRNA modified synthesis. However, our understanding of how the MnmE is involved in plant chloroplast development is scarce, let alone in rice. A novel rice mutant, *thermo-sensitive chlorophyll-deficient mutant 8* (*tcd8*) was identified in this study, which apparently presented an albino phenotype at 20 °C but a normal green over 24 °C, coincided with chloroplast development and chlorophyll content. Map-based cloning and complementary test revealed the *TCD8* encoded a multi-domain GTPase localized in chloroplasts. In addition, the disturbance of *TCD8* suppressed the transcripts of certain chloroplast-related genes at low temperature, although the genes were recoverable to nearly normal levels at high temperature (32 °C), indicating that *TCD8* governs chloroplast development at low temperature. The multi-domain GTPase gene in rice is first reported in this study, which endorses the importance in exploring chloroplast development in rice.

## 1. Introduction

Chloroplast is a core plastid managing plant life activities such as growth and development, photosynthesis, and stress response. In addition, the development of plastids from protoplasts to photosynthetic chloroplasts is a paramount important metabolic process. Clearly, the transcription of chloroplast genes is completed under the co-ordination of PEP and NEP [1,2]. NEP is a phage-type single-subunit RNA polymerase encoded by the nuclear gene *RpoT*, which plays a role in the proplastids and is involved in the transcription; the PEP is a fungal multi-subunit RNA polymerase, being composed by four core subunits of α, β, β′, β″, encoded by the chloroplast genes *rpoA*, *rpoB*, *rpoC1*, and *rpoC2*, respectively, and are responsible for transcription by NEP [3]. It is well known that chloroplast development involved many processes related to chloroplast differentiation which can be divided into three steps [4,5,6]: (i) the activation of plastid replication and plastid DNA synthesis; (ii) the establishment of chloroplasts characterized by the setting up of a chloroplast genetic system. In this step, NEP preferentially transcribes plastid genes encoding their expression machineries, and the transcription and translation activities in chloroplasts are enhanced greatly; (iii) the plastid and nuclear genes that encode the photosynthetic apparatus are then highly expressed. In spite of this, the expression mechanism of many genes involved in chloroplast development in higher plants are still not fully appraised yet [7].

GTPases with enzymatic activity are commonly found in all lifecycles and play a vital role in many cellular processes including biosynthesis and translocation, ribosome assembly, signal transduction, membrane trafficking, and cell cycle control [8]. Those GTPases can be classed into small GTPase and TRAFAC (translation factor), a member of the P-loop GTPase (different from small GTPase) which usually exist in plants [9,10,11] and can be divided into five super-families including the TRAFAC super-family. Additionally, among the TRAFAC super-family [10], there exists a multi-domain GTPase (also known as MnmE), which is highly conserved amongst bacteria and eukarya [12]. It is deemed that the MnmE usually forms a transient complex with MnmG (also known as GidA), which plays a critical role in tRNA modifications [13,14,15]. It is mentionable that 18 GTPases have been identified in *Arabidopsis thaliana*, and there is at least one chloroplast protein in each family/subfamily, suggesting eubacteria-related Era (*E. coli* Ras-like protein)-like GTPases play an important part in chloroplasts [10]. Notably, *Arabidopsis* multi-domain GTPase (MnmE protein) was targeted exclusively to chloroplasts and possibly acted a role in tRNA wobble uridine modification in chloroplasts [10,16]. In spite of all this, our knowledge of the functional mechanism of multi-domain GTPase in higher plants is lagging far behind *Arabidopsis*, let alone in rice.

An effective approach for elucidating the developmental function of plant chloroplast is by using chlorophyll-deficient mutants. In this research, a novel rice mutant *tcd8* that encodes a multi-domain GTPase (MnmE) in rice and shows albino phenotype at 20 °C at seedling stage while being normal green over 24 °C, was discovered. Additionally, the mutation of *TCD8* suppressed the gene transcription related to photosynthesis, chloroplast development, and chlorophyll biosynthesis under low temperature. Our outcomes recommended that *TCD8* plays a significant part in rice chloroplast development at low temperatures.

## 2. Materials and Methods

### 2.1. Materials and Growth Conditions

The *tcd8* mutant was derived from *japonica* rice variety Jiahua 1 (wild-type, WT), treated by ^60^Co gamma-radiation in 2006. The crossing of Pei’ai 64S (*indica*) and the *tcd8* mutant was made to get the F_2_ genetic mapping population. Growth chambers or paddy field were used to grow rice plants under local environmental conditions of Shanghai (31°11 N) and Hainan (18°16 N), in summer and winter season, respectively.

### 2.2. Phenotype Observation and Chlorophyll Determination

Chlorophyll a, b, and carotenoid were determined following Jiang et al. (2014) [17]. To phenotype and pigment analysis, WT and *tcd8* seedlings were continuously grown in growth chambers under four temperatures of 20, 24, 28, and 32 °C, respectively and 12 h dark/12 h light interval with the light intensity of 120 μmol/m^2^·s. The experimental treatments were arranged in three replicates; 200 mg of fresh seedling leaves samples were then collected from each treatment at 3-leaf-stage and incubated at 4 °C for 18 h in dark with 5 mL of extraction buffer (ethanol: acetone: water = 5:4:1)

### 2.3. TEM (Transmission Electron Microscopy) Analysis

The third leaves of WT and *tcd8* of 3-leaf-stage seedlings grown at 20 °C and 32 °C were cut up at about 1 mm^2^ fragments of the middle same part of the leaves. The transmission electron microscope slides were obtained by multiple fixing of the sampled fragment leaves in a solution of 2.5% glutaraldehyde followed by 1% OsO4 buffer and cleaning, and other treatments as well. The slides were examined and photographed by a Hitachi-765 Transmission Electron Microscope (Tokyo, Japan) according to Lin et al. (2021) [18].

### 2.4. TCD8 Gene Mapping

All genomic DNA were extracted from rice seedlings, following the modified CTAB method [19]. Firstly, 130 pairs of single sequence repeats (SSR) and InDel primers were detected using data in Gramene (http://gramene.org/, accessed on 17 May 2019), and screened to determine the chromosomal location of the *TCD8* gene. The novel InDel and SSR markers (Appendix A) were then designed by the Primer 5.0 software (www.PremierBiosoft.com, accessed on 19 October 2022, PREMIER Biosoft International, San Francisco, CA, USA) according to the whole genomic sequences of *japonica* Nipponbare and *indica* 9311 [20,21]. For fine-mapping, a total of 868 F_2_ mutants were collected in this study. In the target region, the predicted function of the candidate gene was acquired via TIGR (http://rice.plantbiology.msu.edu/cgi-bin/gbrowse/rice/, accessed on 19 October 2022).

### 2.5. Complementation Experiment

For complementation study, 5864-bp DNA fragment including 1.2 kb up-stream promoter region sequence, the entire *TCD8* (*LOC_Os08g31460*) coding region as well as 0.6 kb down-stream sequence was magnified from WT plants using the primers, 5′-GAGGTACCCAAGTTTAAGTTCTTTATTTTCCACACG-3′ and 5′-GGGTCGACCGATAATTAATGTTTCTTTAAGAAGATACG-3′. Restriction site (*Kpn*I and *Sal*I) were represented here as underline sequence. The PCR product was cloned into the pMD18-T vector to construct the pMD18-T-TCD8 cloning vector and, post sequence verification, the target fragment was cloned into the pCAMBIA1301 binary vector (CAMBIA; https://cambia.org/, accessed on 19 October 2022). Then, pCAMBIA1301-*TCD8* plasmids were transferred into EHA105 (*Agrobacterium strain)* and introduced into the *tcd8* mutants through *Agrobacterium*-mediated transformation [22]. The genotyping of transgenic plants was defined by PCR amplification of *GUS* marker genes (*GUS* F: 5′-GGGATCCATCGCAGCGTAATG-3′ and *GUS* R: 5′-GCCGACAGCAGCAGTTTCATC-3′) in the expression vector and the DNA of the non-transgenic mutant plants was as a negative control; positive plants were selected for transplantation.

### 2.6. Subcellular Localization of TCD8

The cDNA fragment encoding the N-terminal region of *TCD8* (1–548 amino acids) without the stop codon was amplified from total RNA in WT plants using the primer pairs, 5′-GAAGATCTATGGCTCGCGCTCTCTCTCGCC-3′ (*Bgl*II); 5′-GGGGTACCCGCCGCAAGCACAGGTCATTCC-3′ (*Kpn*I) (the underlined sequences were the cutting site of *Bgl*II and *Kpn*I, respectively). Next, the PCR product was introduced into vector pMON530-GFP. The obtained pMON530-TCD8-GFP plasmids were further introduced into tobacco (*Nicotiana tabacum*) leaves and co-cultured at 28 °C for 48 h using the previous methods in Wang et al. (2017) [23]. Meanwhile, the pMON530-GFP empty carrier was taken as a control. Later, TCD8-GFP fluorescence was examined in tobacco cells using Zeiss confocal laser scanning microscope (LSM 5 PASCAL, http://www.zeiss.com, accessed on 19 October 2022).

### 2.7. RNA Transcript Analysis

Total rice RNA extraction was carried with TRIzol Reagent and DNase I treated RNA using an RNeasy kit, following the instructions of manufacturers. The extracted RNA was then reversed to cDNA by Rever-TraAce (ToYoBo, Osaka, Japan). For expression pattern analysis, the RNA of WT plant tissues including the R (roots), S (stems), and YL (leaves of young-seedling), SL (top second leaf), FL (flag-leaf at heading), and P (young panicles) were extracted. Total RNA of the third fresh leaves of WT and *tcd8* seedlings grown at 20 °C and 32 °C were extracted, respectively, for qRT-PCR analysis. To transcript analysis of the related genes for photosynthesis, chlorophyll biosynthesis, and chloroplast development, 23 genes (*OsActin*, *CAO*, *HEMA, PORA, LhcpII*, *cab1R, YGL1*, *psaA*, *psbA*, *rbcL*, *rbcS*, *OsPOLP*, *FtsZ*, *OsRpoTp*, *OsV4*, *V1*, *V3*(*RNRL*), *rpoA*, *rpoB*, *rpoC*, *TCD5*, *TCD8, TCD10*) were selected, with the rice housekeeping gene, *OsActin* (*LOC_Os03g13170*), as the internal control. qPCR amplifications were conducted in an ABI7500 Real-Time PCR System and the relative quantification of gene expression data were analyzed as described by Livak and Schmittgen [24]. As our previous studies [17], the above primer sequences, gene functions, and references of these genes were tabulated in Appendix A. All experiments were conducted in three biological repeats.

### 2.8. Sequence and Phylogenetic Analysis

Analysis of TCD8-encoded proteins and conserved domain structures were predicted using pfam (http://pfam.xfam.org/search/sequence, accessed on 19 October 2022), and the homologous proteins of TCD8 in different species were found and downloaded by TCD8 protein sequence in Protein BLAST of NCBI website. The amino acid sequence was compared with DNAMAN software, and MEGA7 (http://www.megasoftware.net, accessed on 19 October 2022) was used to construct the phylogenetic tree.

## 3. Results

### 3.1. Phenotypic Characteristics of the tcd8 Mutant

Phenotype of *tcd8* seedlings, grown at 20, 24, 28, and 32 °C, respectively, were shown in Figure 1. Obviously, the *tcd8* seedlings displayed albino phenotype at 20 °C, but revived green over 24 °C, indicating the temperature sensitive of leaf color. Furthermore, the accumulations of pigment contents such as carotenoid and chlorophyll a, b in *tcd8* mutants at 20 °C were much lower than those at 32 °C and those in WT plants (Figure 1b,c), showing that the low temperature restricted the formation of photosynthetic pigments in *tcd8* mutants. In addition, WT cells (Figure 2a,e) presented normal chloroplasts with well-organized lamellar structures that are equipped with normally stacked grana and thylakoid membranes, regardless of temperatures. Whereas, at 20 °C, most *tcd8* mutant cells contained fewer chloroplasts (Figure 2c), and plastids were severely vacuolated (Figure 2d), implying the *tcd8* mutation led to malformed chloroplasts of young seedlings at low temperature.

### 3.2. TCD8 Encodes a Multi-Domain GTPase Protein

The phenotypic segregation (3:1) of green to albino plants in F_2_ population, generated by Peiai64S/*tcd8* (Appendix A), clearly showed the mutant trait of *tcd8* was caused by a single nuclear recessive gene (*tcd8*) mutation. At first, 112 F_2_ mutant individuals were used to locate *TCD8* locus that was between OP08-19i and MM2936 on chromosome 8 (Figure 3a). A total of 868 F_2_ mutant individuals were then used and five InDel molecular markers were developed (Appendix A). Eventually, the *TCD8* was narrowed to 173 kb between InDel markers ID13441 and ID13582 (Figure 3b) on two BAC clones (AP004636.3 and AP00491.2). In this target region, fifteen candidate genes were predicted by program RGAP Project (http://rice.plantbiology.msu.edu/cgi-bin/gbrowse/rice/, accessed on 19 October 2022) (Figure 3c). Of all candidate genes sequenced, only a 1 bp deletion (A) at position 3214 bp from the ATG start codon of *LOC_Os08g31460* functionally encoded a multi-domain GTPase protein (Figure 3d) that resulted in the premature termination of translation in *tcd8* mutants (Appendix A).

A genetic complementation was carried out to further confirm the cause-and-effect relationship between the mutation of *LOC_Os08g31460* and the *tcd8* phenotype. In this study, the differentiation of rice callus was induced at 20 °C to facilitate observation. The results showed that all obtained transgenic T_0_ seedlings harboring pCAMBIA1301-TCD8 presented normal green WT behavior (Appendix A) at 20 °C, whereas non-transgenic seedlings persisted albino phenotype (Appendix A). In addition, the T_1_ progeny showed the segregation of mutant plants at 20 °C (Figure 3e). The observations affirmed that *LOC_Os08g31460* was the *TCD8* gene.

### 3.3. Characterization of TCD8 Protein

*TCD8* is composed of nine exons and nine introns (Figure 3d) and encodes a novel GTPase protein of 570 aa with a molecular weight about 62 kDa. Based on TargetP [25] and the Pfam database [26], TCD8 not only contains a chloroplast transit peptide (CTP; 43 aa), it also embodies a MnmE-N domain from 87 aa to 210 aa and a helical domain of MnmE from 213 aa to 567 aa (Appendix A) which belong to multi-domain GTPase (MnmE). Clearly, the *tcd8* mutation just rightly happened within the MnmE domain (at 505 aa) (Appendix A), harmed the integrity of the function protein, and may also cause the dysfunction of TCD8 protein (Appendix A).

Furthermore, the study showed TCD8 is conserved in higher plants, especially in *Sorghum*, *Brachypodium distachyon*, and *Aegilops tauschii*, with similarities of 81.94, 81.09, and 80.94%, respectively (Appendix A). Phylogenetic analyses demonstrated that the TCD8 homologs were also preserved in different species from bacteria to eukarya (Appendix A).

### 3.4. Subcellular Localization and Expression Patterns of TCD8

TCD8 protein was predictably localized in the chloroplast with TargetP [25]. To ascertain this, the *TCD8* protein was fused to pMON530-CaMV35S-GFP plasmid which was introduced into tobacco cells in a transient expression assay, with the empty vector as a control. The GFP signals from TCD8-GFP were localized to chloroplasts (Figure 4). This confirms that TCD8 is localized at the chloroplasts. In addition, the mRNA transcript levels of *TCD8* by semi-quantitative RT-PCR in various tissues showed that *TCD8* was strongly expressed in leaves, but much weakly expressed in other tissues (Figure 5). This was consistent with the data in the RiceXPro database (Appendix A), suggesting the importance of *TCD8* in chloroplast development in rice.

### 3.5. The Disruption of TCD8 Changes the Transcription Levels of Asosicated Genes

To assess whether the mutation of *TCD8* impairs expression of related genes, the transcript levels of 23 genes for photosynthesis, chlorophyll biosynthesis, and chloroplast development in *tcd8* and WT seedlings, grown at 20 °C and 32 °C, respectively, were investigated. Under low temperature, there were not obvious differences in *HEMA* (glutamyl tRNA reductase) [27], *CAO* (chlorophyllide an oxygenase1) [28], and *rbcL* (the large subunit of Rubisco) [29], but a significant up-regulation in *rbcS* (the small subunit of Rubisco) [30], for those ten genes related to chlorophyll synthesis (Figure 6a) and photosynthesis (Figure 7b) in *tcd8* mutants as compared to WT plants. However, the other six genes (*PORA*, *YGL1*, *cab1R*, *psaA*, *psbA*, *LhcpII*) [31,32] were down-regulated (Figure 6a,b); Among chloroplast-development genes, except for *V1* (the chloroplast-localized protein NUS1) [33] and *TCD5* (a monooxygenase) [34] that had high expressions, all other remaining genes were down-regulated in the mutants, including *tcd8* gene itself (Figure 6c), *YGL1* (a chlorophyll synthetase) [35], *cab1R* (the light harvesting chlorophyll a/b-binding protein of PS II) [30,36], *psaA* (the P700 chlorophyll a apoprotein of PS I) [37], *OsRpoTp* (NEP core subunits) [38], *rpoC* (RNA polymerase beta’ subunit) [3,39]. Notwithstanding, at 32 °C, all the down-regulated genes abovementioned at 20 °C, recovered to normal levels as WT plants (Figure 7a–c), corresponding to the low temperature sensitivity of the mutant phenotype. In conclusion, under low temperature, the transcript level of genes related to chloroplast development, photosynthesis, and chlorophyll biosynthesis reduced due to mutation in *tcd8*.

## 4. Discussion

In this study, a multi-domain GTPase (MnmE) *TCD8,* indispensable for chloroplast development at low temperature in rice, was detected and characterized. Rice plants with dysfunction of *TCD8* produced incomplete chloroplasts and displayed a chlorophyll-deficient phenotype during the seedling stage grown at low temperature, arising from the abnormal expression of genes related to photosynthesis, chlorophyll biosynthesis, and chloroplast development. Our findings revealed the important role of *TCD8* in rice chloroplast development at low temperature.

### 4.1. TCD8 for Chloroplast Development under Low Temperature

The chloroplast, descended from an ancient endosymbiotic cyanobacterium, is a semi-autonomous organelle containing about 100 genes [40]. It is well acknowledged that the chloroplast development essentially requires coordination between plastid and nuclear genes. The mutation of any of these genes may lead to chloroplast defects/chlorophyll deficiency in rice. In rice, about 70 mutants of chlorophyll biosynthesis or chloroplast developmental defects have been determined [41]. Some of them, such as *v1*, *v2*, *v3*, *v5*, *chs1*, *chs2*, *chs3*, *chs4*, and *chs5*, are known to be sensitive to low temperature [42]. Additionally, *TCD9* (α subunit of chaperonin protein 60) [17], *OsV4* (a PPR protein) [43], *WLP1* (the chloroplast ribosome L13 protein, RPL13) [44], *TCD5* (a monooxygenase) [34], *TCD11* (the plastid ribosomal protein S6) [23], *TSV3* (the Obg-like GTPase) [18], *TCM1* (a component of the TAC complex) [45], *OsSIG2A* (RNA polymerase sigma factor) [46], and *TCM12* (the 2, 3-bisphosphoglycerate-independent phosphoglycerate mutase) [47] were identified to be essential for rice chloroplast development at low temperature. Additionally, it was also reported that *RPS15* and *RPL33* in tobacco were encompassed for the development of chloroplast under cold stress [48,49]. In this study, transcript levels of certain genes pertaining to photosynthesis, chlorophyll biosynthesis, and chloroplast development were impaired under low temperatures (20 °C) in the *tcd8* mutants (Figure 6); while with a rising of temperature, the down-regulated genes could be resumed to normal level or even higher than WT levels (Figure 7). This may lead to the alteration in pigment content and chloroplast structure between 20 °C and 32 °C. The observations revealed that *TCD8* is indispensable for chloroplast development under cold stress.

### 4.2. TCD8 May Regulate the First Step of Chloroplast Development in Rice

As aforementioned, the development of plastids from protoplasts to mature chloroplasts can be roughly partitioned into three steps [4,42,50]. The first step involves *OsPOLP* [51,52], *FtsZ* [53,54], and *TCD10* [55]. *OsRpoTp* (NEP), *V2*, and *rpoA* [38,53,56], *OsV4* [43], and *TCD5* were involved in the second step [34]. The third step of chloroplast development involves PEP-transcribed plastid genes (e.g., *cab1R*, *psaA*, *LhcpI*, *psbA*, *rbcL*) [6], were down-regulated in the *tcd8* mutants (Figure 6b). Similarly, *TCD8* also contributes to the second step of chloroplast development because of the low transcript levels for PPR gene *OsV4*, *OsRpoTp*, and the NEP-transcribed gene *rpoB* and *rpoC* involving in this step. More importantly, in view of the fact that the transcripts of three known key genes (*OsPOLP*, *FtsZ*, and *V3)* in the first step were significantly down-regulated (Figure 6c) in *tcd8* mutants, and the inhibition in the first step definitely led to suppression in its second and third steps of chloroplast development, it is then reasonably inferred that *TCD8* may also regulate the first stage of chloroplast development in rice.

Moreover, it is worth mentioning that *Arabidopsis* multi-domain GTPase (MnmE) was reported to localize only in chloroplasts and have a vital role in tRNA modification in chloroplasts [10]. Similarly, the fact that the growth in *Pseudomonas syringae* MnmE mutant was severely hindered at 4 °C, but not at 22 °C and 28 °C affirmed the importance of MnmE at low temperature [57]. Furthermore, MnmE, assisted by its chaperone MnmG, participated in the synthesis of a tRNA wobble uridine modification [16]. As *tcd8* mutation occurred in a highly conserved MnmE domain and led to change of protein structure (Appendix A), it might affect the binding between MnmE and MnmG and, subsequently, its role of tRNA modification in chloroplast at low temperatures. Despite there being TCD8 involvement in rice chloroplast development under low temperature, advance study needs to be conducted to further explore its function.

## 5. Conclusions

In conclusion, exploring the underlying mechanism of chloroplast development at low temperature is essential for improving our understanding of plant growth under cold stress. The mutation of *TCD8* encoding the MnmE protein causes chloroplast development defect at early leaf stage of rice at low temperatures. Our findings evidently revealed that the cold-inducible *TCD8* is important for regulation of cold-responsive genes in rice and also essential for early rice chloroplast development.

## Figures and Tables

**Figure 1 biology-11-01738-f001:**
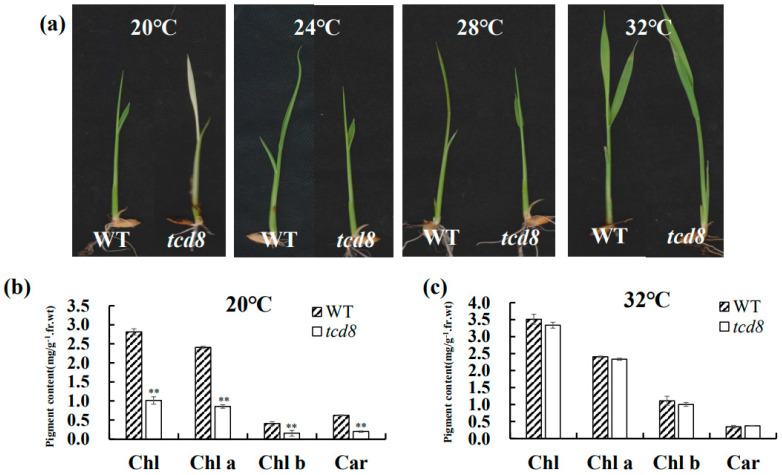
Characterization of the *tcd8* mutants: (**a**) 3-leaf-stage phenotype of (WT (left) and *tcd8* mutant (right) grown at 20 °C, 24 °C, 28 °C, and 32 °C, respectively; (**b**) photosynthetic pigment contents of 3-leaf-stage seedlings at 20 °C, and (**c**) at 32 °C. Chl, total chlorophyll; Chla, chlorophyll a; Chlb, chlorophyll b; Car, carotenoid. Bars represent the mean ± SD of three biological replicates. Raw data were used to conduct Student’s t-test; asterisks indicate statistically significant differences: ** *p* < 0.01.

**Figure 2 biology-11-01738-f002:**
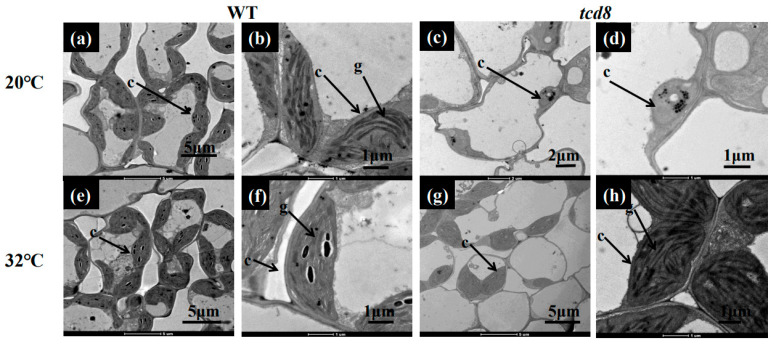
TEM (Transmission electron microscopic) images of chloroplasts in WT and *tcd8* mutant: (**a**) WT cells under 20 °C; (**c**) The cells of *tcd8* mutant under 20 °C; (**b**,**d**) Magnified views of chloroplasts in the (**a**,**c**); (**e**) WT cells under 32 °C; (**g**) *tcd8* cells mutant under 32 °C; (**f**,**h**) Magnified views of chloroplasts in (**e**,**g**). c, chloroplast; g, grana lamella stacks.

**Figure 3 biology-11-01738-f003:**
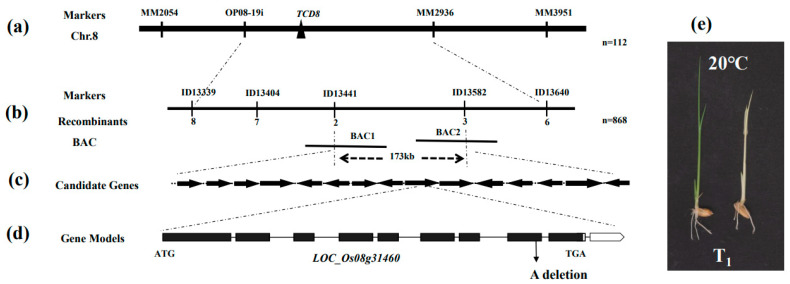
Mapped-cloning of the *TCD8* gene: (**a**) *TCD8* was located between the OP08-19i and MM2936 on chromosome 8 (Chr.8) using 112 F_2_ mutant individuals; (**b**) Fine-mapping of TCD8 between BAC1 (AP004636.3) and BAC2 (AP00491.2) within a 173 kb region by the markers ID13441 and ID13582 using 868 mutant individuals; (**c**) The target region contains fifteen predicted candidate genes (*LOC_Os08g31320*, *LOC_Os08g31340*, *LOC_Os08g31360*, *LOC_Os08g31410*, *LOC_Os08g31420*, *LOC_Os08g31430*, *LOC_Os08g31440*, *LOC_Os08g31450*, *LOC_Os08g31460*, *LOC_Os08g31470*, *LOC_Os08g31510*, *LOC_Os08g31520*, *LOC_Os08g31550*, *LOC_Os08g31560*, *LOC_Os08g31569*); (**d**) Comparison with wild-type gene sequence revealed a single nucleotide (A) deletion mutation at the eighth exon (represented by black squares from left to right). The (A) deletion mutation at the position 3214 bp from the ATG start codon in *LOC_Os08g31460*, and (**e**) Complementation of the *tcd8* mutant, segregation of T_1_ plants obtained from transgenic T_0_ plants.

**Figure 4 biology-11-01738-f004:**
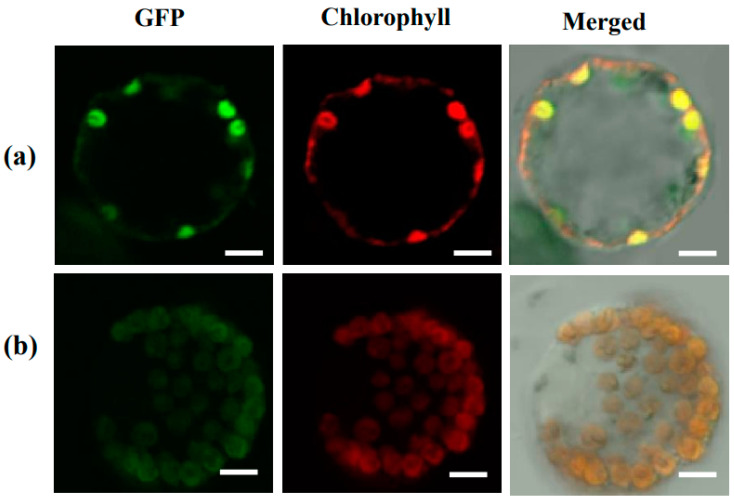
Subcellular localization of *TCD8* gene: (**a**) Empty GFP vector and (**b**) TCD8-GFP fusion protein. The scale bar represents 5 μm.

**Figure 5 biology-11-01738-f005:**
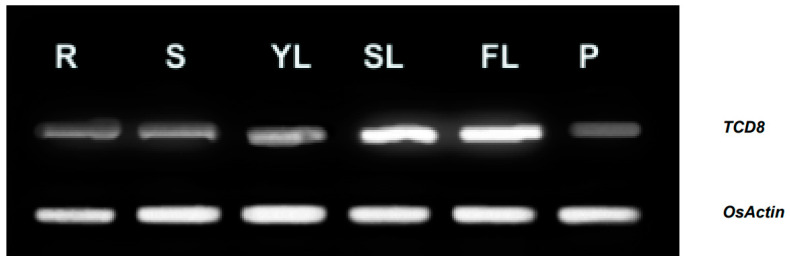
Transcript levels of TCD8 in various tissues. R (young-seedling roots), S (young-seedling stems), YL (young-seedling leaves), SL (second leaf from the top), FL (flag leaf at heading), P (young panicles). *OsActin* was used as a control.

**Figure 6 biology-11-01738-f006:**
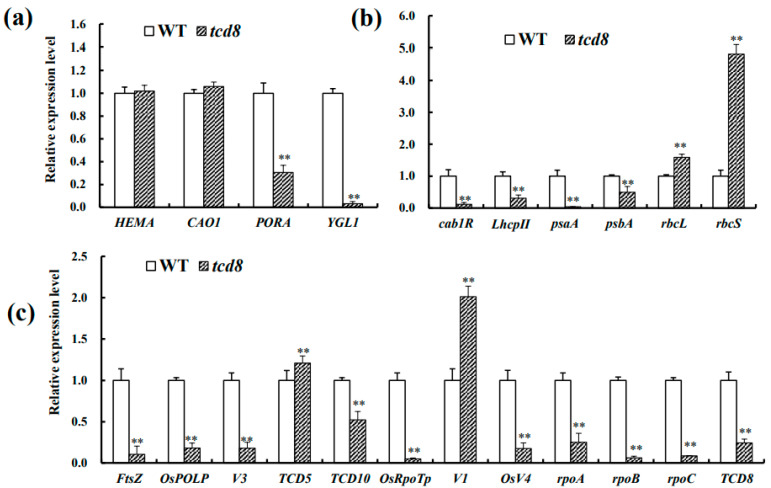
qRT-PCR analysis of related genes in WT and tcd8 mutant at the 3-leaf-stage under 20 °C: (**a**) Analysis of genes associated with chlorophyll biosynthesis; (**b**) Photosynthesis, and (**c**) chloroplast development. *OsActin* was used as an internal control to analyze the relative expression of genes in WT and *tcd8.* The relative expression level of each gene in WT and tcd8 was analyzed by using OsActin as an internal control. Bars represent the mean ± SD of three biological replicates. Raw data were used to conduct Student’s *t-*test; asterisks indicate statistically significant differences: ** *p* < 0.01.

**Figure 7 biology-11-01738-f007:**
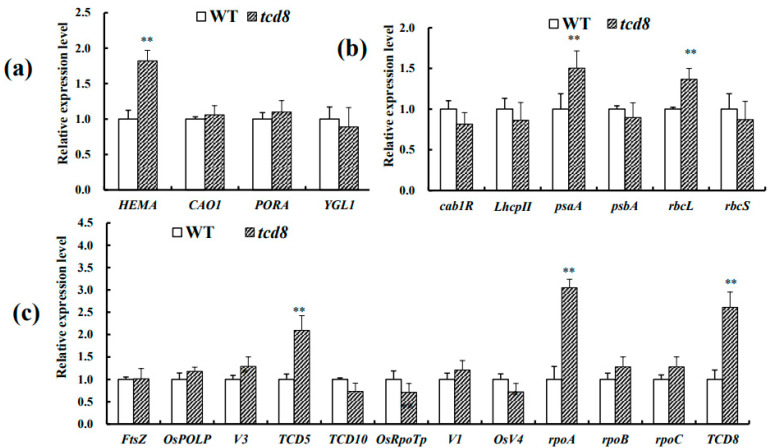
qRT-PCR analysis of related genes in WT and tcd8 mutant at the 3-leaf-stage under 32 °C: (**a**) Analysis of genes associated with chlorophyll biosynthesis; (**b**) photosynthesis, and (**c**) chloroplast development. *OsActin* was used as an internal control to analyze the relative expression genes in WT and *tcd8*. Bars represent the mean ± SD of three biological replicates. Raw data were used to perform Student’s-test; asterisks indicate statistically significant differences: ** *p* < 0.01.

## Data Availability

Not applicable.

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
