# Peer review of "Rice TCD8 Encoding a Multi-Domain GTPase Is Crucial for Chloroplast Development of Early Leaf Stage at Low Temperatures"

_biology, 2022, doi:10.3390/biology11121738_

Round 1

Reviewer 1 Report

This manuscript described a forward genetic study of a thermo-sensitive chlorophyll-deficient rice mutant. Overall, it is well-designed and presented with solid experimental data to identify and characterize the causative gene, even though it falls short of the goal to reveal the functional mechanism of GTPase in chlorophyll biosynthesis and chloroplast development. I have a main comment below which could improve the manuscript.

In addition to the spatial expression in various tissues, the mRNA transcript level of TCD8 under different temperature conditions between the wild-type and the mutant should be studied by qRT-PCR to reveal that TCD8 can be induced by low temperature only in WT. If possible, the expression of TCD8 protein can be experimentally studied as well. The regulatory role of TCD8 on some of the downstream genes also can be explored, at least discussed in the manuscript to reflect the goal of functional mechanism characterization.

Author Response

Thank you for giving us such a high evaluation. We further study funtion of TCD8 in protein level in future.

Reviewer 2 Report

The authors report a rice mutant named thermo-sensitive chlorophyll-deficient mutant 8 (tcd8), which shows albino phenotype at 20 oC. The mutant has less photosynthetic pigments, and chloroplast development in the mutant is impeded. At higher temperatures, however, the mutant is normal. The TCD8 gene is highly expressed in leaves. In the tcd8 mutant, multiple genes that are involved in chloroplast development show abnormal expression levels at 20 oC. Therefore, it is concluded that TCD8 is needed for chloroplast development at low temperature in rice. I found the finding in this paper is interesting and the research can help us to better understand how chloroplast development is regulated.

My comments are listed below.

1.      Writing of the introduction needs to be improved. Some sentences are difficult to understand. As an example,

-Lines 36-37, “...which plays a role in the proplastids, and is involved in the transcription”. Do the authors mean NEP plays a role in transcribing proplastid genes?

-Lines 43-45. “NEP preferentially transcribes plastid genes encoding plastid gene expression machineries and the transcription and translation activities in chloroplasts increased significantly”. I would suggest the authors to break this sentence into two sentences.

-Line 46. “...that encode the photosynthetic apparatus”. Genes do not encode photosynthetic apparatus.

-Line 50. “...many cellular processes including biosynthesis and translocation”. Please specify biosynthesis and translocation of what.

Similar issues can be found in other parts of the introduction and in the discussion. Please check carefully.

2.      Lines 32-33. Proplastids not “protoplasts”.

3.      Figure S1 needs a remake. It is not appropriate to use exact the same figure from another paper.

4.      Figures 1, 6, and 7. “the ± SD”, mean is missing. Add a space between Student’s and t-test. “photosynthetic pigment contents”, make the first p capital size. This issue can be found in other figures too.

5.      Please correct errors of degree symbol like the ones in line 174 throughout the whole paper.

6.      Figure 2. Please check the scale bars to make sure that they are correct.

7.      Figure 3e. This image is not the right image to show the complementation.

8.      Line 198. “Based on the phenotypic separation...”. Should be segregation.

9.      Figure S4. The label TrmE_N in (a) and (b) does not match in the main text where it is referred as MnmE_N. In (a) and (b), the transit peptide should also be labelled. For (c), It should be mentioned in the figure legend or in the main text that the predicted structure of the mutant protein lacks the two helixes, which exist in the WT protein.

10.   Figure S5. The figure title should be protein sequence alignment, not phylogenic analysis.

11.   Figure S6. The tree needs a remake. And this is an unrooted tree. Scale bar cannot be found in this figure.

12.   Figure 4. Why does GFP alone in (a) gives a strong chlorophyll signal. Are the images taken by using leaf protoplasts? If yes, the method of making protoplasts must be included in the methods.

13.   Line 240. “...the TCD8 protein”, do not use font italic for proteins. Please correct this thoroughly in the paper.

14.   I would suggest the authors to include a detailed review of how GTPase proteins regulate chloroplast development. And discuss more about the possible mechanism of how TCD8, as a GTPase, regulates gene expression and chloroplast development.

Author Response

Author Response: Thank you very much for reviewing our paper.

My comments are listed below.

Writing of the introduction needs to be improved. Some sentences are difficult to understand. As an example,

Author Response: Thank you very much for reviewing our paper.

-Lines 36-37, “...which plays a role in the proplastids, and is involved in the transcription”. Do the authors mean NEP plays a role in transcribing proplastid genes?

Author Response: NEP plays a role in transcribing nuclear genes involved in chloroplast development

-Lines 43-45. “NEP preferentially transcribes plastid genes encoding plastid gene expression machineries and the transcription and translation activities in chloroplasts increased significantly”. I would suggest the authors to break this sentence into two sentences.

Author Response: As suggested, we revised it.

-Line 46. “...that encode the photosynthetic apparatus”. Genes do not encode photosynthetic apparatus.

Author Response: Photosynthetic apparatus contain photosynthetic proteins and other components in chloroplasts.

-Line 50. “...many cellular processes including biosynthesis and translocation”. Please specify biosynthesis and translocation of what.

Author Response: Thank you pointing out. Biosynthesis also naturally contain photosynthesis processes etc.

Similar issues can be found in other parts of the introduction and in the discussion. Please check carefully.

Lines 32-33. Proplastids not “protoplasts”.

Author Response: Of course, “Proplastids” is not “protoplasts”.

Figure S1 needs a remake. It is not appropriate to use exact the same figure from another paper.

Author Response: We have stated in the note that we quoted from their original image. In other hand, from respect of the intellectual property rights of creators, using original image is better.

Figures 1, 6, and 7. “the ± SD”, mean is missing. Add a space between Student’s and t-test. “photosynthetic pigment contents”, make the first p capital size. This issue can be found in other figures too.

Author Response: As you suggested, we revised them.

Please correct errors of degree symbol like the ones in line 174 throughout the whole paper.

Author Response: As you suggested, we revised it.

Figure 2. Please check the scale bars to make sure that they are correct.

Author Response: As you suggested, we revised them.

Figure 3e. This image is not the right image to show the complementation.

Author Response: The segregation of T1 plants obtained from transgenic T0 plants transformed with pCAMBIA1301-TCD8 were listed in Figure 3(e). In detail, right rice plants indicate T1 plants, left rice plants, without TCD8 transgene, showed the white seeding.

Line 198. “Based on the phenotypic separation...”. Should be segregation.

Author Response: As you suggested, we corrected it.

Figure S4. The label TrmE_N in (a) and (b) does not match in the main text where it is referred as MnmEN. In (a) and (b), the transit peptide should also be labelled. For (c), It should be mentioned in the figure legend or in the main text that the predicted structure of the mutant protein lacks the two helixes, which exist in the WT protein.

Author Response: As you suggested, we added related sentences of “The predicted structure of the mutant protein lacks the two helixes”in figure legend.

Figure S5. The figure title should be protein sequence alignment, not phylogenic analysis.

Author Response: As you suggested, we corrected it.

Figure S6. The tree needs a remake. And this is an unrooted tree. Scale bar cannot be found in this figure.

Author Response: In order to protect and respect the intellectual property rights of creators, we used the original image. By the way, Scale bar existed in Figure S6.

Figure 4. Why does GFP alone in (a) gives a strong chlorophyll signal. Are the images taken by using leaf protoplasts? If yes, the method of making protoplasts must be included in the methods.

Author Response: Thank you for your criticism. Tobacco (Nicotiana tabacum) leaves was taken for TCD8 subcellular localization using the our previously reported methods in Wang et al. (2017) [23].

Line 240. “...the TCD8 protein”, do not use font italic for proteins. Please correct this thoroughly in the paper.

Author Response: As you suggested, we corrected it.

I would suggest the authors to include a detailed review of how GTPase proteins regulate chloroplast development. And discuss more about the possible mechanism of how TCD8, as a GTPase, regulates gene expression and chloroplast development.

Author Response: Thank you for your criticism. We will continue to carry out more detailed research on the function of rice TCD8 gene in the future.

Reviewer 3 Report

This manuscript reports on the response of rice role for TCD8 encoding the multi-domain GTPase in chloroplast development at low temperatures. The results were clearly presented, however, there are some points needed to be addressed. I will in my review focus on experimental design and manuscript set-up.

- There is a lack of literature review of the two/three past years.

Line 92: change “1mm2” to 1 mm2”.

- Please add accesion number of the genes in the Table S2.

- Conclusion should be expanded more.

Author Response

Author Response: Thank you very much for reviewing our paper and giving us a positive response.

- There is a lack of literature review of the two/three past years.

Author Response: Thank you for your criticism. We have not found the directly related literature of rice TCD8 gene or GTP research within 2 or 3 years.

- Line 92: change “1mm2” to 1 mm2”.

Author Response: As you suggested, we corrected it.
- Please add accesion number of the genes in the Table S2.
Author Response: We believe that with detailed references, accesion number of the genes are not needed

- Conclusion should be expanded more.

Author Response: Thank you for your criticism. We will continue to carry out more detailed research on the function of TCD8 gene in the future.

Round 2

Reviewer 2 Report

Lines 31-32, “...the development of plastids from protoplasts to photosynthetic chloroplasts...”. A protoplast is a treated plant cell that does not have the cell wall. It is a cell. I think the authors mean chloroplast development from proplastids, which are the undifferentiated plastids.

Line 34. When PEP and NEP first appear in the paper, the authors need to provide the full names of these two proteins. Their full names can be found in ref. 2.

Lines 34-38, “NEP is a phage-type single-subunit RNA polymerase encoded by the nuclear gene RpoT, which plays a role in the proplastids and is involved in the transcription; the PEP is a fungal multi-subunit RNA polymerase, being composed by four core subunits of α, β, β’, β’’, encoded by the chloroplast genes rpoA, rpoB, rpoC1, and rpoC2, respectively, and are responsible for transcription by NEP.”. The way how this sentence is written makes me think that NEP transcribes plastid genes. And this is what we know from other literatures. But the authors responded my previous concern about the writing of this sentence with “NEP plays a role in transcribing nuclear genes involved in chloroplast development”. The authors should provide evidence showing that NEP transcribe nuclear genes instead of plastid genes.

Line 44-45, “the genes of nucleus and plastid encode the photosynthetic apparatus are then highly expressed”. Genes encode proteins and RNAs. Proteins and RNAs produce photosynthetic apparatus. It is not appropriate to say that genes encode apparatus.

Figure S1. If the authors insist not to remake this figure, the authors should contact the original authors to get their permission for using their figure without any change in this paper.

I was not able to see the revised supplementary figures. But as the authors mentioned when they responded to my previous comment on figure S6, if this figure was also an exact copy from another publication, and if the authors insist to use it, they should contact the original maker to get a permission.

Figure 4. The authors did not respond to my concern why GFP alone show strong chlorophyll location.

Author Response

Author Response: Thank you very much for reviewing our pape. My comments are listed below.

Lines 31-32, “...the development of plastids from protoplasts to photosynthetic chloroplasts...”. A protoplast is a treated plant cell that does not have the cell wall. It is a cell. I think the authors mean chloroplast development from proplastids, which are the undifferentiated plastids.

Author Response: Thank you for your rigorous and scientific attitude.

Line 34. When PEP and NEP first appear in the paper, the authors need to provide the full names of these two proteins. Their full names can be found in ref. 2.Lines 34-38, “NEP is a phage-type single-subunit RNA polymerase encoded by the nuclear gene RpoT, which plays a role in the proplastids and is involved in the transcription; the PEP is a fungal multi-subunit RNA polymerase, being composed by four core subunits of α, β, β’, β’’, encoded by the chloroplast genes rpoA, rpoB, rpoC1, and rpoC2, respectively, and are responsible for transcription by NEP.”. The way how this sentence is written makes me think that NEP transcribes plastid genes. And this is what we know from other literatures. But the authors responded my previous concern about the writing of this sentence with “NEP plays a role in transcribing nuclear genes involved in chloroplast development”. The authors should provide evidence showing that NEP transcribe nuclear genes instead of plastid genes.

Author Response: As you suggested, we added full names of these two proteins (NEP and PEP).

Line 44-45, “the genes of nucleus and plastid encode the photosynthetic apparatus are then highly expressed”. Genes encode proteins and RNAs. Proteins and RNAs produce photosynthetic apparatus. It is not appropriate to say that genes encode apparatus.

Author Response: As we suggested, we revised it accordingly.

Figure S1. If the authors insist not to remake this figure, the authors should contact the original authors to get their permission for using their figure without any change in this paper. I was not able to see the revised supplementary figures. But as the authors mentioned when they responded to my previous comment on figure S6, if this figure was also an exact copy from another publication, and if the authors insist to use it, they should contact the original maker to get a permission.

Author Response: As you pointed out, we revised them. Since Figure S1 was cited from Reference 10(Suwastika et al., 2014)in original manuscript, the absence of Figure S1 will not affect reading and understanding. Therefore, Figure S1 has been deleted from our revised version. Of course, we have also adjusted the number of Supplementary Figures in the revised manuscript.

Figure 4. The authors did not respond to my concern why GFP alone show strong chlorophyll location.

Author Response: We don't know why. We thiink it will not affect our results.